# Interactions between Platelets and Tumor Microenvironment Components in Ovarian Cancer and Their Implications for Treatment and Clinical Outcomes

**DOI:** 10.3390/cancers15041282

**Published:** 2023-02-17

**Authors:** Selin Oncul, Min Soon Cho

**Affiliations:** Section of Benign Hematology, Department of Pulmonary Medicine, The University of Texas MD Anderson Cancer Center, Houston, TX 77030, USA

**Keywords:** ovarian cancer, platelet, tumor microenvironment, metastasis, angiogenesis, thrombosis, immune system

## Abstract

**Simple Summary:**

Despite initially responding to treatment, many ovarian cancers recur because of tumor cell heterogeneity, chemoresistance, and the cancer-promoting and immunosuppressive tumor microenvironment. Recurrent tumors account for the reduced overall and progression-free survival of patients with ovarian cancer. Ovarian cancer is commonly accompanied by thrombocytosis and thrombotic events, which implies that platelets may participate in cancer progression via their association with cancer cells and the tumor microenvironment. This review focuses on platelets’ interactions with cellular and acellular components of the tumor microenvironment, including endothelial cells, mesenchymal stem cells, adipocytes, pericytes, immune cells, and extracellular matrix elements, and discusses how these interactions support the proliferation and metastasis of ovarian cancer cells. It also provides an overview of potential therapeutic strategies that obstruct platelets’ protumor effects by reprogramming the tumor microenvironment.

**Abstract:**

Platelets, the primary operatives of hemostasis that contribute to blood coagulation and wound healing after blood vessel injury, are also involved in pathological conditions, including cancer. Malignancy-associated thrombosis is common in ovarian cancer patients and is associated with poor clinical outcomes. Platelets extravasate into the tumor microenvironment in ovarian cancer and interact with cancer cells and non-cancerous elements. Ovarian cancer cells also activate platelets. The communication between activated platelets, cancer cells, and the tumor microenvironment is via various platelet membrane proteins or mediators released through degranulation or the secretion of microvesicles from platelets. These interactions trigger signaling cascades in tumors that promote ovarian cancer progression, metastasis, and neoangiogenesis. This review discusses how interactions between platelets, cancer cells, cancer stem cells, stromal cells, and the extracellular matrix in the tumor microenvironment influence ovarian cancer progression. It also presents novel potential therapeutic approaches toward this gynecological cancer.

## 1. Introduction

Ovarian cancer is the fifth most common cause of death among women and the most lethal gynecological malignancy in the United States [1]. Epithelial ovarian cancer, which accounts for 90% of ovarian cancers, is further categorized into serous, endometrioid, mucinous, and clear cell types, in addition to other rare or non-specified subgroups [2]. The most prevalent type of epithelial ovarian cancer, high-grade serous ovarian cancer (HGSOC), originates in the fallopian tubes or ovarian surface epithelium and disseminates to the ovaries and peritoneum [3]. HGSOC, which carries a poor prognosis, is often aggressive, diagnosed in its late stages, and has a high capacity for metastasis and ascites formation [2]. Non-epithelial ovarian cancers (e.g., germ cell and sex cord–stromal) constitute approximately 10% of all ovarian cancers and are generally less aggressive than epithelial ovarian cancer [4]. Risk factors for ovarian cancer include genetic alterations, including germline mutations of breast cancer genes 1 and 2 (*BRCA1* and *BRCA2*) [5] and alternative mutations of DNA damage repair-associated genes such as *BRIP1*, *RAD51*, and *ATM*/*ATR* [6,7]; Lynch syndrome [8]; and various environmental factors such as hormone replacement therapy [9], not giving birth and/or not breastfeeding [10], being overweight [11], and frequent tobacco smoking [12]. The conventional therapy for ovarian cancer is surgical excision of the tumor along with neoadjuvant or adjuvant platinum- or taxane-based chemotherapy [13]. Additional treatments include using agents that target proangiogenic factors, poly (ADP-ribose) polymerase (PARP), and immune system components [3].

Despite marked improvements in the efficacy of treatments, ovarian cancer is associated with high mortality because of late diagnosis due to its anatomical location and mostly indistinguishable symptoms, high recurrence rate [2], primary or acquired resistance to chemotherapy [3,4], and tumor heterogeneity [5]. Thus, more specific and effective approaches are in demand to inhibit tumor growth and prolong patient progression-free survival and overall survival. Because the tumor microenvironment (TME) is a principal contributor to cancer progression and poor therapeutic response [6,7], targeting TME components holds excellent potential for more effective cancer management.

The TME comprises cancer cells; cancer stem cells (CSCs); stromal cells such as fibroblasts; endothelial cells; immune cells; and proteins of the extracellular matrix (ECM) such as collagen and fibronectin [8]. Crosstalk among these components constitutes a complex signaling network that promotes malignancy and metastasis [9]. Accumulating evidence indicates that TME has a role in the development and progression of ovarian cancer [10,11], and many studies manipulate TME in the treatment of ovarian cancer [12,13].

One of the components of TME in ovarian cancer is extravasated platelets [14,15]. Membrane proteins on activated platelets, including P-selectin [16], GPIbα [17], GPIIb/IIIa [18], and C-type lectin-like receptor 2 (CLEC-2) [19], mediate the binding of platelets to various elements in the TME. The released contents of granules from activated platelets, such as growth hormones [20], cytokines [21], and chemokines [22], also affect the TME. Additionally, platelet microparticles [23], mitochondria [24], and nucleic acids [25,26] can reshape the TME response in cancer. The major molecules on platelets or in their granules are listed in Table 1. We also summarize the protumorigenic and antitumorigenic influences of various platelet-associated molecules in the TME in Table 2. The interactions between platelets and the TME components of ovarian cancer are illustrated in Figure 1.

Healthy individuals have platelet counts of between 150,000 and 450,000 per microliter of blood. In contrast, roughly one-third of newly diagnosed ovarian cancer patients have platelet counts exceeding 450,000 per microliter [27]. In patients with ovarian cancer, thrombocytosis is an adverse prognostic factor associated with elevated serum carcinoma antigen 125 (CA-125) levels, advanced disease stage, and poor clinical outcomes [27,28], as well as the diminished efficacy of secondary cytoreductive surgery [29] and chemotherapy [30]. Aside from contributing to the formation of venous thromboembolisms, platelets contribute to cancer progression via distinct mechanisms, including increasing proliferation [31], epithelial–mesenchymal transition (EMT) [32], and anoikis resistance in cancer cells [33]; promoting the formation of the premetastatic niche and metastasis [33]; enhancing angiogenesis [27] and the integrity of tumor vasculature [34]; inducing immune tolerance [35]; and reducing the impact of chemotherapy [30].

**Table 1 cancers-15-01282-t001:** The major components reside on activated platelets or are released from platelet granules.

Location	General Function	Examples	References
**Surface** **molecules**	Integrins	α2β1 (GPIa/IIa), α5β1, α6β1, αLβ2 (ICAM-2), αIIbβ3 (GPIIb/IIIa), αVβ3	[36]
	Selectins	P-selectin (CD62P), CLEC-2	[37,38]
	Leucine-rich repeat receptors	GPIb-IX-V, TLR1, TLR2, TLR4, TLR6, MMPs	[36,39,40]
	ADP receptors	P2Y_1_, P2Y_12_	[41]
	Thrombin receptors	PAR1, PAR4, GPIbα	[42,43]
	Tetraspanins	CD63, CD9, CD53	[36]
	Prostaglandin receptors	PGD2 and PGE2 receptors	[44]
	Prostacyclin receptors	PGI2 receptors	[44]
	Thromboxane receptors	TxA2 receptors	[45]
	Lipid receptors	PAF and LPA receptors	[46,47]
	Ig receptors	GPVI, FcγRIIA (CD32), FcεRI (CD23)	[48,49]
	JAMs	JAM-1, JAM-2, JAM-3, PECAM-1 (CD31)	[50,51]
	Tyrosine kinase receptors	Thrombopoietin, leptin, insulin, PDGF receptors	[36]
	Immune checkpoints	PD-L1, GITRL, OX40L	[52,53,54]
	Other receptors	Serotonin receptors, GPIV (CD36), IAP (CD47) complement receptors, CD40, CD40L (CD154)	[55,56,57,58,59]
**α-granules**	Adhesion molecules	vWF, αIIbβ3 (GPIIb/IIIa), αVβ3, P-selectin (CD62P), fibrinogen, fibronectin, thrombospondin	[36,60,61]
	Proangiogenic factors	VEGF, angiopoietin-1, SDF-1 (CXCL12), S1P, TGF-β, IL-6, PF4 (CXCL4)	[61]
	Angiostatic factors	Endostatin, angiostatin, thrombospondin-1	[56,61]
	Growth factors	VEGF, PDGF, EGF, FGF, HGF, IGF-1, CTGF, TGF-β	[61,62,63,64]
	Coagulation-associated components	Prothrombin, fibrinogen, factor V, factor VIII, factor XI, protein S	[61,65,66]
	Fibrinolytic factors	α2-macroglobulin, uPA, PAI-1	[61]
	MMPs	MMP-1, MMP-2, MMP-3, MMP-9	[67,68]
	Metalloproteinases	ADAM-10, ADAM-17, ADAMTS-13	[69]
	TIMPs	TIMP-1, TIMP-2, TIMP-4	[69]
	Inflammamodulatory molecules	CXCL1, PF4 (CXCL4), CXCL5, CXCL7 (NAP-2), IL-1β, IL-6, IL-8 (CXCL8), SDF-1 (CXCL12), CCL2 (MCP-1), CCL3 (MIP-1α), CCL5 (RANTES), CCL7, PAF, LPA, TGF-β, TNF-α, GM-CSF	[61,70,71,72,73,74,75]
	Immunologic molecules	Complement factors, IgG, IgA, IgM, thymosin-β4	[61,76,77,78]
	Other components	Albumin, α1-antitrypsin, HMWK	[61]
**δ-granules**	Nucleotides	ADP, ATP, GDP, GTP	[79]
	Bioactive amines	Serotonin, histamine, epinephrine	[60]
	Ions	Calcium, magnesium, phosphate, pyrophosphate	[61]
	Polyphosphates	Polyphosphate (polyP)	[60]
**Lysosomes**	Proteases	Cathepsin D/E, carboxypeptidase A/B, glycohydrolases, collagenase, elastase	[79,80]
	Phosphatases	Acid phosphatase	[79]
	Phospholipases	Phospholipase A	[61]

ADAM: a disintegrin and metalloproteinase; CCL: chemokine (C-C motif) ligand; CD40L: CD40 ligand; CLEC-2: C-type lectin-like receptor 2; CTGF: connective tissue growth factor; CXCL: chemokine (C-X-C motif) ligand; EGF: epidermal growth factor; FGF: fibroblast growth factor; GITRL: glucocorticoid-induced tumor necrosis factor receptor-related protein ligand; GM-CSF: granulocyte-monocyte colony-stimulating factor; GP: glycoprotein; HGF: hepatocyte growth factor; HMWK: high-molecular-weight kininogen; IAP: integrin-associated protein; ICAM-2: intercellular adhesion molecule 2; Ig: immunoglobulin; IGF-1: insulin-like growth factor 1; IL: interleukin; JAM: junctional adhesion molecule; LPA: lysophosphatidic acid; MCP-1: monocyte chemoattractant protein 1; MIP: migration inhibitory protein; MMP: matrix metalloproteinase; NAP-2: neutrophil-activating peptide 2; PAF: platelet-activating factor; PAI-1: plasminogen activator inhibitor 1; PAR: protease-activated receptor; PD-L1: programmed death ligand 1; PDGF: platelet-derived growth factor; PECAM-1: platelet endothelial cell adhesion molecule 1; PF4: platelet factor 4; PGD2: prostaglandin D2; PGE2: prostaglandin E2; PGI2: prostaglandin I2; polyP: polyphosphate; RANTES: regulated upon activation and normal T cell expressed and secreted; S1P: sphingosine 1-phosphate; SDF-1: stromal-derived factor 1; TGF-β: transforming growth factor-beta; TIMP: tissue inhibitor of metalloproteinases; TLR: Toll-like receptor; TNF-α: tumor necrosis factor-alpha; TxA2: thromboxane A2; uPA: urokinase-type plasminogen; VEGF: vascular endothelial growth factor; vWF: von Willebrand factor.

This review focuses on the interactions between platelets and TME components, the contribution of these interactions to the progression of ovarian cancer, and the plausible approaches to interrupting this interwork to improve the prognosis of patients with ovarian cancer.

## 2. Interactions of Platelets with TME Compartments: Endothelial Cells, Pericytes, and Cancer-Associated Fibroblasts

### 2.1. Interactions with Endothelial Cells

#### 2.1.1. In Angiogenesis

Tumor angiogenesis involves degradation of the vascular endothelial matrix, the proliferation and migration of endothelial cells, the branching of endothelial cells to generate vascular rings, and the establishment of new basement membranes [81]. Tumor blood vessels, which tend to be erratic, branched, and leaky, are dissimilar to normal blood vessels in terms of shape, integrity, and permeability. Moreover, perivascular cells are reduced in number and are less likely to be associated with endothelial cells [82,83]. Blood flow in tumor-associated vessels is inconsistent and may lead to maladjusted circulation [82,84]. Consequently, tumors cannot receive adequate oxygen and nutrients, and discharge excess carbon dioxide and other metabolites generated by the glycolytic pathway. The TME becomes more hypoxic, acidic, and ischemic [85]. In addition, the hyperpermeability of the tumor vasculature enhances extravascular clotting, fibrin gel clot formation, and endothelial and stromal cell expansion [86]. Angiogenesis is a poor prognostic factor in ovarian cancer [87], and antiangiogenic therapeutics demonstrate a moderate effect on overall and progression-free survival [88,89].

Platelets preferentially attach to tumor-associated vessels rather than normal vasculature, amplifying the delivery of tumorigenic mediators to the TME [90]. Tumor cell-induced platelet activation (TCIPA) leads to the translocation of P-selectin (also known as CD62P), a cell adhesion molecule stored in the α-granules [37], to the platelet surface. The binding of P-selectin to the P-selectin glycoprotein ligand (PSGL-1) on leukocytes governs leukocyte rolling in activated endothelial cells [91] and the generation of platelet—cancer cell complexes [92]. Adhesion molecules, including integrins, von Willebrand factor (vWF), fibrinogen, fibronectin, and coagulation factors, and several members of the a disintegrin and metalloproteinase (ADAM) protein family accommodate the activation, tethering, rolling, and firm adhesion of platelets to endothelial cells [93]. Activated platelets degranulate and release various factors that affect angiogenesis. More than 30 components associated with platelets that influence angiogenesis have been described [94]. Platelets generate angiostatic factors such as endostatin, angiostatin, and thrombospondin-1 (TSP-1), and angiogenic factors including vascular endothelial growth factor (VEGF), angiopoietin-1, stromal-derived factor 1 (SDF-1, also known as the chemokine (C-X-C motif) ligand [CXCL]12), sphingosine 1-phosphate (S1P), transforming growth factor-beta (TGF-β), interleukin (IL)-6, and platelet factor 4 (PF4; also known as CXCL4) [61]. Platelet-derived growth factor (PDGF) supports the function of cancer-associated fibroblasts (CAFs), vascular pericytes, and smooth muscle cells in angiogenesis [95]. Platelets also support the recruitment of endothelial progenitor cells (EPCs) [96]. Platelet integrin GPIIb/IIIa promotes endothelial cell proliferation and function [97]. The activation of platelets and the release of their granular content, such as angiopoietin-1 and serotonin, prevent intratumoral bleeding [98]. ATP released from the δ-granules of platelets activates endothelial P2Y_2_ receptors, causing the retraction of endothelial cells and promoting the transendothelial migration of cancer cells (intravasation and extravasation) and metastasis [99]. Platelet microparticles increase the expression of matrix metalloproteinases (MMPs) on endothelial cells [100], assisting in the generation of new vessels [61].

The co-localization of GPIIb (CD41), platelet endothelial cell adhesion molecule-1 (PECAM-1; also known as CD31), and VEGF in ovarian cancer tissues suggests the involvement of platelets in angiogenesis and tumor growth [101]. An increased level of VEGF can be considered a biomarker of ovarian cancer [102] and an indicator of advanced disease, ascites formation, metastasis, and reduced survival [103,104]. Moreover, the levels of PDGF-BB and VEGF were found to be positively correlated in the TME and ascites, and the pharmacological inhibition of their receptors increased the efficacy of chemotherapy in patients with ovarian cancer [105]. The co-localization of regulator of G-protein signaling 5 (RGS5), a signal transduction molecule upregulated in endothelial cells in the TME, with PECAM-1 and PDGF receptor (PDGFR)-β, has been reported in various types of cancer, including ovarian cancer [106]. The participation of activated platelets in angiogenesis is displayed in Figure 2.

Platelet GPIb-IX receptor complex, a receptor for vWF, and GPIIb/IIIa, the receptor for fibrinogen, promote platelet aggregation [107] and adhesion to endothelial cells. Additionally, platelet P-selectin and GPIIb assist in the adhesion of platelets to cancer cells. Hence, platelets assist in the clinging of cancer cells to the endothelium and metastasis [108,109]. Targeting platelet surface proteins might show therapeutic benefits in cancer. Antiplatelet agent-directed platelet inhibition diminishes the proliferative capability of ovarian cancer cells [31]. In addition, focal adhesion kinase (FAK) promotes platelet infiltration into the TME, and targeting FAK suppresses ovarian tumor growth. Dual therapy using antiplatelet agents and antiangiogenic drugs prevents rebound tumor growth after discontinuing antiangiogenic agents [20].

#### 2.1.2. In Lymphangiogenesis

Lymphangiogenesis is the formation of new lymphatic vessels and occurs during embryonic development and in pathological conditions involving inflammation and tumor metastasis [110]. Platelets are essential for the proper partitioning of blood and lymphatic vessels during development, mainly by coordinating endothelial cells’ expansion, relocation, and tube formation. This phenomenon occurs following the engagement of platelet CLEC-2 with its ligand podoplanin on lymphatic endothelial cells [111]. Although platelets do not necessarily contribute to the maintenance of the separation of the two circulatory systems post-development in normal conditions, in certain situations, including wound healing or tumor growth, platelets again participate in lymphangiogenesis [112]. Platelets stimulate lymphangiogenesis by secreting proangiogenic factors such as VEGF, angiopoietin-1, PDGF, and insulin-like growth factor 1 (IGF-1) [61,113], and through the interaction of CLEC-2 and podoplanin [111].

In patients with ovarian cancer, the upregulation of lymphangiogenic markers is associated with more aggressive disease and shorter overall survival [114]. Podoplanin overexpression in the malignant stroma of ovarian cancer patients predicts lymphatic spread and poor clinical outcomes [115]. Blocking podoplanin—CLEC-2 contact between ovarian cancer cells and platelets forestalls lymph vessel proliferation [116]. Likewise, VEGF and PDGF released by platelets promote lymph vessel generation in epithelial ovarian cancer [117]. Treatment with antiangiogenic agents might attenuate lymphangiogenesis in ovarian cancer [118]. Notably, inhibiting the TGF-β signaling cascade prevents lymphangiogenesis and subsequent VEGF-mediated ascites generation in ovarian cancer patients [119].

### 2.2. Interactions with Pericytes

Pericytes are perivascular cells embedded in the basement membrane surrounding the microvasculature [120]. Pericytes might prevent the intravasation of cancer cells and metastasis [121]; however, they might also facilitate micrometastasis by supporting the formation of tumor vasculature [122]. Tumor vessels have atypical coverage of pericytes, whose contact with endothelial cells is disrupted [122]. The tumor vasculature has an excess of pericytes that loosely interact with endothelial cells, deteriorating the integrity of the vessels and resulting in hemorrhage [123]. In ovarian cancer, abnormal pericyte numbers and expression signatures are associated with tumor growth, aggressive metastasis, and poor clinical outcomes [124].

Podoplanin, which is highly expressed on pericytes, mediates platelet binding to pericytes via CLEC-2 [125]. Furthermore, platelet-derived TGF-β, angiopoietin, and PDGF also stimulate pericyte differentiation, colonization, and their interaction with endothelial cells [64]. TGF-β strongly influences the proliferation of pericytes [126] and their association with endothelial cells through the TGF-β—matrix protein axis [127]. The activation of the TGF-β signaling cascade impacts the density and lumen size of tumor microvessels [127]. Angiopoietin overexpression is linked with pericyte impairment and tumor vessel instability [128]. Blocking TGF-β or angiopoietin signaling inhibits tumor growth and neovascularization [127,129]. PDGF released from platelets and other cells [130] is essential for pericytes recruitment and function during tumor neoangiogenesis. Preventing PDGF isoforms from binding to their receptors and hindering angiogenesis with antiangiogenic agents such as bevacizumab may interfere with the incorporation of pericytes into new blood vessels [131].

TSP-1 is a matricellular glycoprotein with antiangiogenic properties that counters the proliferative effects of growth factors on endothelial cells [132,133]. TSP-1 is released from the granules of activated platelets, prompting platelet aggregation and tethering [56]. In ovarian cancer, binding of the TSP-1 three type 1 repeats (3TSR) domain of TSP-1 to GPIV (CD36) normalizes the tumor vasculature and exhibits antitumor function [134]. Treating patients with 3TSR in combination with chemotherapeutics [135] or oncolytic viruses [136] can improve the efficacy of anticancer therapies. 3TSR alone, or fused with the Fc region of human IgG1 for improved stability, increases the number of pericyte-covered blood vessels, reduces the proliferative capacity of endothelial cells, and contributes to vascular normalization [134].

Platelets release IL-6 [73] and also trigger IL-6 secretion from tumor cells by releasing several factors, such as lysophosphatidic acid (LPA) [137]. The protumorigenic cytokine IL-6 is significantly elevated in ovarian cancer patients with confirmed thrombocytosis [27]. High IL-6 levels promote neoangiogenesis with abnormal pericyte coating. Anti-VEGF and anti-IL-6 agents reduce vessel sprouting and leakiness of the vasculature by reinstating the pericyte lining [138]. Combining chemotherapeutic agents and pazopanib, a multitargeted tyrosine kinase inhibitor, can help restore pericyte coverage and restrict tumor microvessel density [139] in patients with ovarian cancer. The antiplatelet action of pazopanib [140] might further inhibit tumor growth and angiogenesis [20].

### 2.3. Interactions with Cancer-Associated Fibroblasts

Fibroblasts are a heterogeneous population of connective tissue cells with a presumably mesenchymal origin [141]. Fibroblasts can differentiate into particular mesenchymal cell types, including osteoblasts, adipocytes, and chondrocytes [142]. In the TME, CAFs promote disease progression by releasing various molecules; rearranging the ECM to facilitate cancer cell motility, invasion, and EMT; and stimulating angiogenesis, tumor growth, and metastasis. CAFs modulate the function of immune cells and the metabolism of cancer cells to promote tumor survival [143]. CAFs also release extracellular vesicles that support cancer progression [144] and chemoresistance [145]. Although fibroblasts play a role in tumorigenesis, they may also restrict tumor development by activating the tumoricidal immune response or consolidating the ECM to prevent tumor dissemination [146].

Extravasated platelets promote EMT by releasing mediators such as TGF-β, SDF-1, and PDGF. The same mediators also induce the differentiation, migration, and proliferation of CAFs [147,148]. PDGF and TGF-β induce mesenchymal stem cell (MSC) differentiation into CAFs [149]. Integrin α11 is a CAF marker, and its expression is related to myofibroblast differentiation and ECM alteration. CAFs expressing integrin α11 and PDGFR-β are associated with poor clinical outcomes in ovarian cancers and other malignancies [150]. Platelet-originated CLEC-2 induces the migration and proliferation of CAFs in the TME [15]. The binding of platelet-derived CLEC-2 to podoplanin on CAFs and cancer cells promote tumor growth and venous thrombosis in patients with ovarian cancer [125,151]. LPA derived from ovarian cancer cells promotes the differentiation of fibroblasts into CAFs through a hypoxia-inducible factor 1 alpha (HIF-1α)-dependent mechanism [152]. LPA activates platelets [153], which, in turn, release LPA [137]. Moreover, LPA promotes the secretion of VEGF and SDF-1 from MSCs, further supporting ovarian cancer progression [154]. Under oxidative stress, platelets release their mitochondria, which are picked up by MSCs [24]. Mitochondria originating in platelets and engulfed by MSCs promote fatty acid synthesis and ATP production and stimulate the release of angiogenic components, such as VEGF and hepatocyte growth factor (HGF), from MSCs [24].

CAFs originating from MSCs release platelet-activating factor (PAF), promoting platelet activation and aggregation [155], which further supports ovarian cancer progression and induces ovarian cancer development through the PAF/PAF receptor signaling pathway [34]. Exosomes from ovarian cancer cells induce the generation of CAFs from MSCs in the tumor stroma [156]. CAF-released IL-6 causes EMT in ovarian cancer cells, tumor growth, and ECM reorganization, mainly by mediating STAT3 phosphorylation [157]. The increased concentrations of IL-6 in the stroma or ascites can activate platelet function and aggregation and lead to thrombosis [158].

Most patients with metastatic ovarian cancer have peritoneal dissemination, which indicates a poor prognosis. It starts with the emigration of cancer cells into the peritoneal fluid, forming floating masses that attach to peritoneal mesothelial cells throughout the peritoneal cavity [159]. Alternatively, the cancer cells can initiate an inflammatory reaction in the peritoneal stroma, promoting the generation of a fibrin mesh that can be used for adhesion to the peritoneal surface. Fibrin mesh can also potentiate the colonization of fibroblasts and endothelial cells. Fibroblasts that differentiate into CAFs promote ovarian cancer invasion through the upregulation of several markers such as alpha-smooth muscle actin (α-MA), PDGFR, and podoplanin [151]. A subset of CAFs originating from mesothelial cells through mesothelial-to-mesenchymal transition (MMT) contribute to peritoneal metastasis [160]. Ovarian cancer cells that have been relocated to the peritoneal cavity promote ascites, and the ascitic fluid contains various cytokines and growth factors [161]. TGF-β derived from ascites and activated platelets is one of the main stimulating factors for MMT [160,162]. In addition, tissue factor (TF), present in high amounts in ascites, cancer cell masses, and cancer cell-derived microparticles, induces thrombin generation, which activates platelets and produces fibrin [163]. Activated platelets further increase the expression of TF, prompting ovarian cancer migration [164]. The activation of mesothelial cells by TGF-β released from platelets is partially responsible for ECM remodeling during metastasis [165].

Like fibroblasts, mesothelial cells and adipocytes in the omentum and peritoneum can be prompted by cancer cells to differentiate into CAFs [166]. Periostin is a secretory protein that is overexpressed by stromal fibroblasts in multiple cancers, including ovarian cancer, and its overexpression is associated with poor clinical outcomes in patients with epithelial ovarian cancer. TGF-β modulates periostin expression and promotes ovarian cancer growth and chemotherapy resistance [167]. Similarly, the aberrant expression and release of connective tissue growth factor (CTGF), a stromal factor, induces the colonization and peritoneal adhesion of ovarian cancer cells [168]. Platelets store a large quantity of CTGF [63], suggesting that platelet activation and the release of CTGF may participate in ovarian cancer seeding in the peritoneum.

Inhibitors of the TGF-β signaling pathway can diminish CAFs’ function in ovarian cancer [169]. Anti-VEGF therapy can inhibit ovarian cancer progression, metastasis, and malignant ascites formation promoted by the release of VEGF from CAFs, along with various other cell types [170]; however, synchronous anti-PDGF treatment might be necessary to target CAFs resistant to VEGF-neutralizing agents [171]. It has been reported that aspirin therapy suppresses chemotherapy-induced CAF formation in colorectal cancer [172] and the impact of CAFs on ovarian cancer.

## 3. Interplay of Platelets with the Tumor Immune Microenvironment

### 3.1. Interplay with Tumor-Associated Neutrophils

Most inflammatory cells in solid tumors are tumor-associated neutrophils (TANs) that promote or inhibit tumor growth and angiogenesis, depending on various circumstances. In the initial stages of tumor growth, TANs tend to exert tumoricidal functions (N1 neutrophils), whereas in later stages of tumor growth, they become protumorigenic (N2 neutrophils). Permissive N2 neutrophils enhance the proliferation and metastasis of malignant cells by releasing growth factors (e.g., TGF-β and VEGF), MMPs, and reactive oxygen species (ROS) [173]. In addition, they can suppress antitumor immune responses [174]. Neutrophils assist cancer cells in escaping the antitumor immune response through the overexpression of immune checkpoint molecules, including programmed death ligand 1 (PD-L1) [175], glucocorticoid-induced tumor necrosis factor receptor-related protein-ligand (GITRL) [176], and V-domain Ig suppressor of T cell activation (VISTA) [177]. In patients with ovarian cancer, elevated neutrophil counts, neutrophil-lymphocyte ratios (NLRs), and platelet counts indicate a poor prognosis and shorter overall survival [178,179].

Adhesion molecules on the surface of neutrophils include L-selectin (CD62L), PSGL-1, macrophage-1 antigen (Mac-1; also known as CD11b/CD18), and leukocyte function-associated antigen 1 (LFA-1; also known as CD11a/CD18), which are essential for the rolling of neutrophils on the endothelium, as well as their firm adhesion and transendothelial migration [180]. Inflammation promotes the expression of surface adhesion molecules on platelets and endothelial cells. Platelets help recruit neutrophils to the tissues via direct or indirect interaction. Neutrophils and platelets interact mainly through the binding of P-selectin to PSGL-1 [181]. The binding of GPIb on platelets to Mac-1 [182] on neutrophils, and that of the intercellular adhesion molecule 2 (ICAM-2) on endothelial cells to LFA-1 [183] on neutrophils, provide additional adhesive interactions under flow conditions. At baseline, Mac-1 has low expression on neutrophils in the peripheral blood; however, it is significantly upregulated in inflammation [184]. Neutrophils in the peripheral blood of ovarian cancer patients have higher levels of Mac-1.

In ovarian cancer patients, neutrophils overexpress Mac-1, which improves the adhesion of neutrophils to endothelial cells and modifies the endothelium for improved cancer cell migration [185]. Of note, the PSGL-1–L-selectin interaction is also important for neutrophil adherence to endothelium [186]. The interaction between cancer cells and neutrophils through Mac-1 and L-selectin can promote cancer cell migration through blood vessels [184,187]. Other interactions, including those between CD40–CD40 ligands (CD40L, also known as CD154) [188] and between platelet PSGL-1 and neutrophil L-selectin [189], might also support neutrophil interactions in the TME.

Platelets and neutrophils communicate indirectly through intermediary molecules. For instance, platelet GPIIb/IIIa and neutrophil Mac-1 both bind to fibrinogen [190]. In addition, platelets’ secretion of chemokines and cytokines, such as PF4 [70], CXCL7 (also known as neutrophil-activating peptide 2 (NAP-2)) [71], and TGF-β [191], following their interaction with cancer cells reinforces neutrophils’ attachment to the endothelium and their transendothelial migration. In ovarian cancer, IL-8 (CXCL8) recruits neutrophils to the TME [192]. IL-8 released from the activated platelets [193] or platelet-activated cancer cells [194] can help increase the number of neutrophils in the TME. IL-8 receptor CXCR2 is upregulated in the neutrophils of patients with ovarian cancer compared to healthy individuals [195]. Moreover, the elevated concentration of IL-8 in peritoneal lavage in ovarian cancer is associated with shorter overall survival [196]. Cytokines and chemokines, such as granulocyte colony-stimulating factor (G-CSF), granulocyte-monocyte colony-stimulating factor (GM-CSF), CXCL1, CXCL2 (also known as migration inhibitory protein 2 alpha (MIP-2α)), CXCL5, and CCL3 (chemokine (C-C motif) ligand 3) (MIP-1α) recruit neutrophils to the TME [197]. TGF-β and G-CSF are important for neutrophil differentiation into the N2 phenotype [173]; thus, platelet-derived TGF-β [198] and GM-CSF [75] create an immunotolerant environment for ovarian cancer.

Neutrophil extracellular trap (NET) formation (NETosis) is a defense mechanism against inflammation [199]. The activated neutrophils secrete antimicrobial proteases and release chromatins and cytosolic and granular proteins, such as myeloperoxidase and neutrophil elastase, to trap and kill pathogens [200]. The presence of lipopolysaccharide (LPS) activates platelets in endotoxemia, and hyperactive platelets bind to adherent neutrophils via Toll-like receptor (TLR) 4, causing neutrophil activation and NETosis. LPS is not adept at directly stimulating NET formation [40]. In return, NETs trigger thrombosis because histone proteins and DNA released from neutrophils, in addition to NET-linked cathepsin G, activate platelets [201]. NETs bind to factor XII or P-selectin to initiate blood coagulation [202]. Neutrophil and the NET-dependent release of neutrophil elastase cleaves E-cadherin and enhances EMT in ovarian cancer cells [203]. Cathepsin G and neutrophil elastase activate platelets via GPIIb/IIIa and induce platelet aggregation in the presence of exogenous fibrinogen [204].

Mitochondrial DNA (mtDNA) and formylated peptides are among the mitochondrial damage-associated molecular patterns discharged from various cells. These molecules stimulate neutrophil function via the adhesion of TLR9 and formylated peptide receptors and promote NETosis [205]. NETs support cancer-related thrombosis through factors such as extracellular chromatin and tissue factor [206]. NETs also play a role in ovarian cancer cell invasion and premetastatic niche formation in the omentum [207]. The amount of platelet-derived mtDNA and the number of microparticles in ascites correlate with poor clinical outcomes in ovarian cancer patients [208]. The fibrin deposits in the TME further assist in ovarian cancer invasion by providing an adhesion surface for platelets, promoting their aggregation, and stimulating monocyte differentiation into tumor-associated macrophage (TAM)-like cells [209]. The pharmacological suppression of NETosis-related genes, including the protein arginine deiminase 4 (*PAD4)* gene, which encodes a histone protein-citrullinating enzyme, can delay ovarian cancer cell invasion of the peritoneal cavity [207].

### 3.2. Interplay with Tumor-Associated Macrophages

Mononuclear cells extravasate into tumor tissues and differentiate into macrophages. Macrophages polarize into two distinct subtypes that have pro- or anti-inflammatory characteristics. Classically activated macrophages, also termed M1 macrophages, are stimulated by molecules such as LPS, interferon-gamma (IFN-γ), tumor necrosis factor-alpha (TNF-α), and GM-CSF, as well as by target pathogens and tumor cells. In contrast, alternatively activated macrophages (M2 macrophages) are induced by molecules such as TGF-β, IL-10, and macrophage colony-stimulating factor (M-CSF), and display protumorigenic and immunomodulatory features [210].

In the ovarian cancer TME, a population of TAMs from both the M1 and M2 phenotypes is observed [211]. As cancer progresses, TAMs polarize into M2 macrophages via the upregulation of multiple markers, including CD163, CD206, PD-L1, and arginase 1 (ARG1) [210]. epithelial ovarian cancers support M2 polarization by releasing mediators such as M-CSF [212]. The number of TAMs escalates as the tumor spreads, so much so that half of the cells in the peritoneal TME and ascites can be TAMs with tolerogenic functions [213,214]. M2 macrophages in ascites induce the transcoelomic metastasis of ovarian cancer cells. M2 macrophages secrete multiple mediators that induce ovarian cancer spheroid formation, encourage attachment to the metastasis site, and escape from the cytolytic immune response [215,216]. An elevated M2/M1 macrophage ratio correlates with reduced overall survival in ovarian cancer patients [217].

Platelet—monocyte aggregates are more potent markers for platelet activation than the expression of P-selectin [218]. The activated macrophages release cytokines such as TNF-α, IL-1β, IL-6, and IL-8 [219]. The interaction between monocytes and platelets enhances the procoagulant effect of macrophages and ROS generation. Activated platelets adhere to monocytes via P-selectin and promote monocyte function and ROS release [220]. A high concentration of ROS differentiates TAMs to the M2 phenotype, and this effect is annihilated when macrophages are polarized to the M1-like phenotype, rather than M2 [221].

The upregulation of platelet-derived TGF-β, the subsequent overexpression of cyclooxygenase 2 (COX-2), and the generation of prostaglandin E2 (PGE2) in monocytes modulate CD16 levels. The activation of CD16 on monocytes induces M2 polarization [222]. COX-2-positive TAMs increase cancer cell proliferation and metastasis because the PGE2 generated by COX-2 activates the protein kinase C/A and TGF-β signaling pathways [223]. COX-2 expression is elevated in macrophages that infiltrate epithelial ovarian cancer tumors [224], and the resulting increased levels of PGE2 support tolerogenic M2 polarization and ovarian cancer cell chemoresistance [225].

The platelet-directed differentiation of monocytes into the immunogenic phenotype via IL-1β and IL-8 potentiates NF-κB signaling and supports thrombosis [226]. Macrophages with high levels of podoplanin expression migrate toward vascular endothelial cells, bind to CLEC-2 on the surface of platelets, activate platelets [38,227], and promote thrombosis [38,227]. Additionally, podoplanin is upregulated in ovarian cancer cells, and extracellular vesicles released from these cells increase the likelihood of thrombosis [125].

Platelet microparticles regulate the function of macrophages recruited to the inflammation site by chemotactic components such as PF4, TGF-β, and PDGF-β [23]. These microparticles can drive macrophages toward the protumorigenic phenotype by suppressing TNF-α, CCL4, and M-CSF production and increasing their phagocytic efficacy [228]. Conversely, platelet microparticles may direct macrophage activation, which can inhibit tumor growth through cytokines such as TNF-related apoptosis-inducing ligand (TRAIL), CCL2 (also known as monocyte chemoattractant protein 1 (MCP-1)) and IL-8 [229].

Cancer cells and activated monocytes can release extracellular vesicles containing TF that contribute to venous thromboembolism [163,230]. Moreover, the coincubation of platelets with extracellular vesicles upregulates the TF activity of monocytes [231]. Cancer cell-derived extracellular vesicles also trigger platelet activation and cause the release of procoagulant platelet microparticles [232]. The extracellular vesicles of platelets with phosphatidylserine attach to monocytes and monocyte extracellular vesicles and promote their procoagulant effects [233,234]. TAMs express the coagulation factors II, V, VII, and X, stimulating thrombin formation and platelet activation [235]. The TF-dependent thrombin formation promotes several cellular interactions in the TME [236]. For instance, the TF-related activation of protease-activated receptor (PAR)-2 and integrin ligation enhance tumor growth and angiogenesis [237]. Moreover, TF supports the intravasation and intravascular survival of cancer cells, and the formation of metastatic niches, to provide an environment for the interaction of monocytes and macrophages with tumor-associated microthrombi [238]. The thrombin receptors PAR1, PAR4, and GPIbα on the surface of platelets can facilitate metastasis [42,43]. By disrupting the TF-accompanying pathways, PAR inhibitors might diminish the proliferation and metastasis of ovarian cancer cells [237].

The network between platelets, cancer cells, endothelial cells, and smooth muscle cells attracts monocytes into the early metastatic niches through chemokines such as CCL2 and CCL5 (known as regulated on activation, normal T cell expressed and secreted (RANTES)) [238]. The CCL5-induced attraction of monocytes to metastatic cancer cells indicates that CCL5 is indirectly critical to the organization of metastasis rather than directly acting on cancer cells [239]. Similarly, platelets’ secretion of SDF-1 is decisive in the recruitment of macrophages to tumors and in the migration of CXCR4^+^ cancer cells [240]. In addition, SDF-1 mediates the migration of TAMs into the hypoxic TME and the polarization of monocytes toward the tolerogenic phenotype [241]. The blockade of the SDF-1—CXCR4 interaction, combined with anti-programmed death 1 (PD-1)-based immunotherapy, results in a higher M1/M2 ratio, which is associated with better clinical outcomes in patients with ovarian cancer [242].

PAFR activation stimulates macrophages to differentiate toward the M2 phenotype, whereas the absence of PAFR drives tumor-infiltrating macrophages to demonstrate antitumor properties [243]. PAFR ligands support tumor development by reprogramming the TAM phenotype and increasing the frequency of M2 macrophages rather than M1 macrophages. This specific polarization of macrophages through apoptotic cells results in a tolerogenic environment that supports tumor growth [243]. Thus, PAFR antagonists may be a therapeutic option for patients with ovarian cancer [244].

Ovarian cancer cell-derived M-CSF stimulates the polarization of macrophages to the M2 subtype, which supports EMT and the peritoneal metastasis of ovarian cancer cells [216]. M-CSF can also trigger platelet aggregation [245] and provoke thrombosis in cancer patients. M2 macrophages release high concentrations of epidermal growth factor (EGF) and induce angiogenesis and the invasion and metastasis of cancer cells through the activation of an EGF receptor (EGFR) [246]. EGF stimulation facilitates the activation of platelets and the release of IL-1β [72]. Moreover, EGF-triggered ovarian cancer cells secrete PAF and contribute to the hyper-responsiveness of platelets. The blockade of EGFR might abolish EGF’s promotion of ovarian cancer progression [247].

In the ovarian cancer TME, LPA is produced from phospholipids by TAM- and platelet-derived phospholipase A2 (PLA2) and autotaxin [248,249]. LPA promotes ovarian cancer growth and metastasis through diverse networks [250,251]. It also induces platelet activation [252]; thus, the pharmacological inhibition of LPA and the application of antiplatelet agents might interfere with ovarian cancer development [249,253].

### 3.3. Interplay with T Cells

#### 3.3.1. CD4^+^ Helper T Cells

CD4^+^ helper T (Th) cells are major contributors to adaptive immunity. Their antitumor effect can be exerted directly by targeting tumor cells or indirectly by triggering the expansion of cytotoxic CD8^+^ T cells [254]. An increased percentage of tumor-infiltrating lymphocytes (TILs) is associated with prolonged survival in patients with ovarian cancer [255,256].

Platelets may affect CD4^+^ T cells directly through physical interaction [257] or indirectly through the secretion of soluble mediators [258], extracellular vesicles, and mitochondria [259,260]. Activated platelets form complexes with CD4^+^ T cells and mediate the adhesion of lymphocytes to the endothelium under flow conditions, and to the ECM [261]. T cells cannot bind to fibronectin efficiently without the participation of platelets [262]. CD40–CD40L, P-selectin–PSGL-1, and ICAM-2–LFA-1 interactions are partly responsible for the construction of platelet–CD4^+^ T cell complexes and the rolling and firm adhesion of T cells under flow [262]. CD4^+^ T cells’ CD40L binds to platelets’ CD40 [58] and stimulates CCL5 secretion [263]. The soluble CD40L (sCD40L) binds to CD40 on CD4^+^ T cells, causing CD4^+^ T cells to undergo apoptosis following the suppressed expression of IL-2, Bcl-2, and Bcl-xL [264]. Platelets, rather than T cells, are the predominant source of sCD40L in cancer [59]. Antiplatelet reagents might restore the antitumor immune response by reducing sCD40L levels and preventing CD4^+^ T cell apoptosis [59].

The interaction between platelet P-selectin and T cell PSGL-1 recruits T cells to the activated endothelium at the site of inflammation or injury. Memory CD4^+^ T cells form complexes with platelets, which may exacerbate thrombo-inflammation [265]. In conjunction with its role in mediating cell adhesion [266], the upregulation of PSGL-1 on effector T cells may alter their function. PSGL-1 engagement can induce T cell exhaustion by inhibiting ERK/Akt signaling and IL-2 secretion, and increasing PD-1 expression. In this context, PSGL-1 ligation can reduce the antitumor effect of CD4^+^ and CD8^+^ T cells [267].

Platelets and CD4^+^ T cells form complexes that can trigger immune-related adverse events in cancer patients treated with chemotherapeutic agents or immune checkpoint inhibitors. The elevated levels of PD-L1 in platelets can inhibit CD4^+^ and CD8^+^ T cells [52]. Moreover, platelet PD-L1 may increase the overall expression of PD-L1 in tumors after platelet adhesion to cancer cells [268], suggesting that adding an antiplatelet agent can improve the efficacy of anti-PD-L1 immunotherapy [269]. On the other hand, other studies show that platelets induce the expression of PD-L1 in cancer cells [14,270], and reducing platelet counts via an antiplatelet agent treatment can minimize the effectiveness of immunotherapy [14]. In addition, platelet-derived TGF-β reduces the efficacy of T cell recruitment via bispecific antibody (BsAb)-based immunotherapy [271] in ovarian cancer [272]. Immune checkpoint inhibitors administered with VEGF inhibitors have been shown to improve the efficacy of BsAbs in ovarian cancer [273].

#### 3.3.2. Regulatory T Cells

Regulatory T cells (Tregs), which constitute 5% to 7% of CD4^+^ T cells, induce immune tolerance by cytokines such as TGF-β and IL-10, and receptors such as cytotoxic T lymphocyte antigen 4 (CTLA-4), CD39, and CD73 [274]. Consequently, Tregs exert protumor effects by providing a permissive environment for tumor growth. In the early stages of ovarian cancer, the Th17-related immune response is more dominant, whereas a persistent shift toward Tregs is witnessed in the later stages [275]. Tregs are recruited and activated by the hypoxic conditions in ovarian cancer TME [276] and are abundant in ascites. Their presence has been linked to unfavorable clinical outcomes [277,278]. Meanwhile, high CD8^+^ T cell/Treg and CD4^+^ T cell/Treg ratios are indicators of a better prognosis [279]. Interestingly, a positive correlation between increased numbers of CD8^+^ T cells and Tregs in the TME has been observed in multiple patients with ovarian cancer, but this immune profile has not influenced patients’ overall survival [280].

Treg-derived PSGL-1 is pivotal in regulating the immune response, since this glycoprotein plays a part in hampering T cell–DC conjugation and subsequent T cell expansion and activation [281]. In contrast, P-selectin is upregulated in CD4^+^ non-Tregs and is critical for Treg differentiation and activity because it functions in signaling cascades involving Syk kinases and CCL5 [263,282,283]. P-selectin–PSGL-1 binding might potentiate the influence of costimulatory components such as CD40L [284]. The engagement of CD40 with CD40L is important for the development and survival of Tregs [285], and CD40–CD40L activates Tregs and platelets in a positive feedback loop [263]. CD40 or CD40L deficiency reduces the number of Tregs without affecting the total number of T cells [286].

Moreover, P-selectin plays an important role in recruiting Tregs to tumors and repressing effector CD8^+^ T cells [16]. IL-4 stimulates endothelial cells’ expression of P-selectin and the recruitment of activated platelets expressing P-selectins to the TME, which causes positive feedback in immunosuppression [287].

Platelet releasate stimulates Treg activity, which, in the short-term, increases and subsequently inhibits Th1/Th17 responses. The mediators secreted by platelets (e.g., PF4 and TGF-β) and by platelet-activated Tregs downregulate Th1 polarization [288]. Platelet deficiency leads to reduced Treg activation through several pathways related to TLR4 and TNF-α. The adherence of platelet-derived TNF-α to TNFR2 on Tregs activates these cells [74]. In the tumor-associated ascites of patients with ovarian cancer, IL-6 stimulates Tregs’ expression of TNFR2 [289]. IL-6 generated by platelets [73] supports the function of Tregs in ascites, creating an immunosuppressive environment. TNFR antagonists might restrict Treg expansion in the TME and hinder ovarian cancer cell proliferation [290].

TGF-β is required to generate Tregs, and platelets, as an important source of TGF-β, are known to promote Treg activity. Platelet TGF-β modulates the expression of granzyme B, IFN-γ, and IL-2 to promote Treg development [291]. Thus, platelet hyperactivity in malignancies may create a favorable environment for Treg expansion [292]. Moreover, platelet-derived growth factors, including IGF-1 [62] and PDGF [293], also foster Treg proliferation. The inhibition of the TGF-β-directed network can mitigate an increased number of Tregs and support cytotoxic T cell expansion while reducing tumor and ascites volumes in patients with ovarian cancer [294]. Similarly, the blockade of IGF [295] and PDGF [296] receptors might restore immunosurveillance for malignant ovarian cells.

PF4 suppresses the overall expansion of CD4^+^ T cells and Th1 polarization while promoting the proliferation of Tregs. The binding of PF4 to its receptor, CXCR3, escalates mitochondrial transcription factor A (TFAM) expression, mitochondrial biogenesis, and ATP and ROS generation. The increased concentrations of ATP and ROS increase T-bet and Foxp3 expression, whichmodulates Th1 and Treg differentiation, respectively [297]. Moreover, thromboxane A2 (TxA2) secreted from platelets inhibits the activity of CD4^+^ T cells [298]. PF4 also heterodimerizes with other platelet-derived chemokines, including CCL2, CCL5, CXCL7, SDF-1, and TxA2. The chemokine pairing aggravates chronic inflammation in cancer; as a result, preventing chemokine pairing might provide a therapeutic advantage [299,300].

CXCR4 and SDF-1 are upregulated in ovarian cancer cells [301]. Platelets express CXCR4 and SDF-1, and activated platelets generate SDF-1 [302,303]. Hypoxia increases the expression of CXCR4 on Tregs and supports immunosuppression [304]. Blocking the binding of SDF-1 to its receptor, CXCR4, increases the CD8^+^ T cell/Treg ratio and reduces the intraperitoneal metastasis of ovarian cancer cells [305]. Tregs with high immune checkpoint expression support the immunoregulatory function of these cells in ovarian cancer [306]. The Treg^hi^ phenotype is distinguished by elevated expression levels of forkhead box P3 (Foxp3), PD-1, 4-1BB, inducible T cell co-stimulator (ICOS), and CD25. The infiltration of these Tregs into tumors is associated with intercepted antitumor immune response [306]. Ovarian tumors display the Treg^hi^ phenotype [278,306], which may explain ovarian cancer patients’ unsatisfactory responses to immune checkpoint inhibitors. In many cancers, PD-L1-containing platelet microparticles adhere to PD-1 on Tregs and induce immune tolerance [307]. The simultaneous inhibition of the CXCR4—SDF-1 and PD-1—PD-L1 pathways and the depletion of effector Tregs may enhance the antitumor immune response and reduce mortality in ovarian cancer patients [242,308].

Platelet microparticles prevent Treg differentiation to proinflammatory T cells that generate IL-17 and IFN-γ [309]. This mechanism occurs when the P-selectin on microparticles adheres to PSGL-1 on CCR6^+^HLA-DR^+^ memory-like Tregs, which are the progenitors of Th17-like cells and stabilizers of the inflammatory state [309,310]. The reduction in IFN-γ release after PF4–CXCR3 engagement [311] and in communication among Tregs upon the binding of platelet microparticles [312] support this mechanism.

#### 3.3.3. CD8^+^ T Cells

CD8^+^ T cells, also termed cytotoxic or cytolytic T lymphocytes (CTLs), play a cardinal role in protecting the host from infections. Unlike acute infections, chronic inflammation and cancer may involve CD8^+^ T cells. CD8^+^ T cells can become exhausted and develop reduced functionality. Exhausted CD8^+^ T cells are distinguished by their persistent overexpression of immune checkpoints [313]. Functional cytotoxic TILs improve survival in ovarian cancer patients [314], and the recruitment of CD8^+^ T cells into the peritoneal cavity is associated with better prognostic scores in these patients [315].

Platelets interact with CD8^+^ T cells and decrease their cytolytic impact on cancer cells. The abundance and size of platelet—T cell aggregates in cancer patients are increased compared to healthy individuals. High levels of these aggregates are regarded as cancer biomarkers [316]. Platelets in platelet—T cell aggregates express high levels of P-selectin, and platelets may be more prone to adhering to PSGL-1 on CD8^+^ T cells than on the surface of CD4^+^ T cells [316]. PSGL-1 is involved in the migration of T cells to the inflammation site and secondary lymphoid tissues [317]. However, in chronic inflammation, PSGL-1 signaling limits T cell survival and promotes CD8^+^ T cell exhaustion because of the upregulation of immune checkpoint molecules such as PD-1, B- and T-lymphocyte attenuator (BTLA), T cell immunoglobulin mucin 3 (TIM-3), and lymphocyte activation gene 3 (LAG-3), and the inhibition of immunostimulatory cytokines such as IL-2. The suppression of PSGL-1 may prevent the exhaustion of cytotoxic T cells and reduce tumor growth [267]. However, this effect may occur not through P-selectin—PSGL-1 ligation but through TCR signaling [267] or the binding to other ligands [318].

TGF-β suppresses CD8^+^ T cells and can promote cancer cell proliferation [319] and metastasis [320]. The activation of latent TGF-β through the TGF-β/glycoprotein A repetitions predominant (GARP) axis in the TME via platelets may inhibit the antitumor effect of T cells [321]. The activation of platelets encourages tumor growth, at least partly, by supporting the immunosuppressive TME and minimizing the adaptive immune response. Therefore, the inhibition of platelets may increase the effects of immune checkpoint inhibitors [322]. In addition, bioengineered PD-1-expressing or antineoplastic agent-carrying platelets may accumulate in tumors and block PD-L1 on cancer cells to help exhausted CD8^+^ T cells recover their functionality [323]. CD8^+^ T Tregs, which are known to reduce the expansion of CD4^+^ T cells and follicular helper T cells, are enriched in ovarian tumors [324]. Indeed, CD8^+^CD25^+^Foxp3^+^CTLA-4^+^ T cells have been detected in patients with ovarian cancer and have increased in number as the tumor stage increases [324,325]. The increased levels of TGF-β expand CD8^+^Tregs in the ovarian cancer TME, mainly through p38 MAPK. The inhibition of TGF-β and the p38 signaling cascade can impede ovarian cancer progression [326]. In addition, various other mediators, including IFN-γ and IL-2, support the regulatory functions of CD8^+^ T cells [324]. In contrast, TGF-β upregulates CD103 (also known as integrin αEβ7 (ITGAE)) on CD8^+^ T cells infiltrating the tumor and malignant ascites, and by supporting the habitation of these cytotoxic T cells in the TME, they may represent a prognostic advantage in patients with ovarian cancer [327].

Thrombin exerts its protumorigenic properties through platelet activation and the subsequent release of various mediators involving VEGF and TGF-β [198,328]. These mediators can further modify the immune response toward a tolerogenic state [198,329]. Specific drugs, such as dabigatran etexilate, when given concurrently with potentially thrombogenic antineoplastic agents, repress thrombin and thus benefit patients with ovarian cancer. This treatment regimen may reduce the number of immunoregulatory myeloid cells and incite CD8^+^ cytotoxic T cell function in ascites [330].

Similarly, platelet-derived PF4 stimulates the differentiation of monocytes into myeloid-derived suppressor cells (MDSCs), which suppresses the activity of CD8^+^ T cells. This shift can result in advanced metastatic disease and poor overall survival [21]. An antiplatelet agent-based therapy regimen can promote the expansion of CD8^+^ T cells, and the administration of antiplatelet agents with an anti-VEGF or anti-PD-1 antibody can resensitize tumors to immune cells [331,332].

### 3.4. Interplay with B Cells

B cells are immune cells that produce antibodies during initial exposure to antigens and memory cells and that mediate a subsequent antibody response after re-exposure to the same antigens. Mature naïve B lymphocytes in the peripheral lymphoid structure are exposed to antigens, selectively expand, hypermutate to generate antibodies that better fit specific antigens, and differentiate into plasma cells or transform into memory B cells [333]. Antibodies produced by B cells may recognize antigens on malignant cells or other components of the TME, tag cancer cells for destruction, and lead to immune cell- and antibody-mediated cytotoxicity [334]. On the other hand, cancer cells or their extracellular vesicles modify the immune system, promote the generation of regulatory B cells (Bregs) [335], and evade the antineoplastic impact of B cells [336]. Moreover, circulating antigen–antibody complexes activate the complement system and platelets, supporting angiogenesis and metastasis [337,338,339]. B cells may have prognostic significance in ovarian cancer patients and have been shown to correlate with worse [340] or better [341] clinical outcomes. High numbers of IL-10-positive Bregs have been detected in ascites of ovarian cancer patients, and their numbers are correlated with the percentage of Tregs. The recruitment of Bregs and Tregs is associated with advanced-stage and aggressive ovarian cancer [335].

The binding of CD40L to CD40 on resting B cells stimulates their differentiation to plasma cells and memory B cells. Platelets and platelet microparticles express CD40L and can trigger B cells’ production of antibodies, although not at the same level as the T cell-dependent isotype-switched antibodies [342]. Through CD40L, platelets enable B cell isotype conversion and enhance germinal center formation [342]. B cells stimulated through CD40 and TLR9 restrict the differentiation of monocytes into mature dendritic cells (DCs) and diminish DC-dependent T cell proliferation and function [343]. CD40–CD40L contact is linked to the activation of Bregs and tumor growth via the suppression of IL-10 and TGF-β1 release [344]. In contrast, CD40L released from B cells binds platelet CD40 and activates platelets, causing the expression of P-selectin, the secretion of α- and γ-granules, the transformation of platelet morphology, and the upregulation of GPIIb/IIIa [58].

In addition to platelets and Tregs, peripheral B cells also express GARP, which releases TGF-β and can induce immune tolerance [345]. TGF-β signaling stimulates the apoptosis of B cells [345], suppresses immunoglobulin release, and impedes surface immunoglobulin expression on activated B lymphocytes [346]. In addition, TGF-β promotes the class switch to IgA during the differentiation of B cells into plasma cells [347]. Although IgA is primarily associated with defense against pathogens on mucosal surfaces, IgG is involved in adaptive and memory immune responses. Accordingly, the trend toward IgA instead of IgG leads to a curtailed cytotoxic immune response in tissue [348]. Bregs’ production of excess TGF-β can further propel the tolerogenic immune response by hastening the differentiation of cytotoxic T cells into Tregs [349]. Thus, platelet-derived TGF-β may contribute to immunosuppressive conditions in tumor progression.

PAF, an inflammatory phospholipid produced mainly by platelets, immune cells, and endothelial cells [350], plays a role in B cell activation, expansion, and Ig production. PAF receptor is upregulated on B cells by several cytokines, including IL-4 and TGF-β. The binding of PAF to its receptors on B cells activates these cells [351].

The adhesion molecules on platelets that support the interaction between platelets and immune cells are summarized in Figure 3.

**Table 2 cancers-15-01282-t002:** The influence of various platelet-associated molecules on TME in cancer.

Hallmark	Platelet Constituents	Prognostic Role	References
**Pro-** **tumorigenic**	Endostatin, angiostatin, TSP-1, angiopoietin-1, VEGF, PDFG, HGF, FGF, SDF-1, S1P, TGF-β, IL-6	Stimulation of angiogenesis	[61,90]
	TGF-β, angiopoietin, PDGF, CLEC-2, IL-6	Proliferation, differentiation, and irregularity of pericytes	[64,125,126,128,138]
	Platelet microparticles	Overexpression of MMPs on endothelial cells	[100]
	P-selectin, GPIIb	Transendothelial migration of cancer cells	[108,109]
	VEGF, angiopoietin-1, PDGF, IGF-1, CLEC-2	Stimulation of lymphangiogenesis	[61,111,113]
	TGF-β, SDF-1, PDGF, CLEC-2	Stimulation and differentiation, proliferation, and migration of CAFs	[15,147,148,149]
	TGF-β, G-CSF	Differentiation of neutrophils into N2-like phenotype	[173]
	IL-8, G-CSF, GM-CSF, CXCL1, CXCL2, CXCL5, MIP-1α	Recruitment of neutrophils into TME	[192,197]
	factor XII, P-selectin, GPIIb/IIIa	Platelet interaction with NETs and subsequent platelet aggregation	[202,204]
	mtDNA, P-selectin	NETosis	[202,205]
	P-selectin, TGF-β, COX-2, PGE2, PF4, PAF, PDGF	Activation of TAMs	[220,221,222]
	CCL2, CCL5, RANTES, SDF-1	Migration of TAMs into TME	[238,241]
	CD40, CCL5, TxA2	Apoptosis and inhibition of CD4^+^ T helper cells	[264,298]
	PF4, TGF-β, P-selectin	Suppression of Th1 and Th17 differentiation	[288,297,309,310]
	P-selectin	Stimulation of T cell exhaustion	[267]
	PD-L1, TGF-β	Inhibition of CD4^+^ and CD8^+^ T cell antitumor functions	[52,319,320,321,326]
	CD40L, P-selectin, TNF-α, IL-6, IGF-1, PDGF, SDF-1	Activation, proliferation, and migration of Tregs	[16,62,73,74,263,285,293,302,303]
	PF4	Differentiation of monocytes into MDSCs	[21]
	CD40L	Expansion of Bregs	[344]
	Platelet microparticles	Overexpression of MMPs	[61,100]
	CXCL5, CXCL7, TGF-β, TSP-1, P2Y_12_, COX-1, TxA2	Formation of pre-metastatic niches	[352]
**Anti-** **tumorigenic**	Endostatin, angiostatin, TSP-1	Inhibition of angiogenesis	[56,61]
	TRAIL, CCL2, MCP-1, IL-8	Activation of tumoricidal macrophages	[229]
	Platelet microparticles	Enhancement of the phagocytic capacity of macrophages	[228]
	TGF-β, EGF	Upregulation of PD-1 in cancer cells and increase in the effectiveness of immunotherapy	[14,270]
	TGF-β, CD40L	Activation and infiltration of CD8^+^ T cells into the tumor	[327,353]
	CD40L	Differentiation of resting B cells into plasma cells and generation of antibodies	[342]
	OX40L	Infiltration of immune cells into TME	[54]

CCL: chemokine (C-C motif) ligand; CD40L: CD40 ligand; CLEC-2: C-type lectin-like receptor 2; COX: cyclooxygenase; CXCL: chemokine (C-X-C motif) ligand; EGF: epidermal growth factor; FGF: fibroblast growth factor; G-CSF: granulocyte colony-stimulating factor; GM-CSF: granulocyte-monocyte colony-stimulating factor; HGF: hepatocyte growth factor; IGF-1: insulin-like growth factor 1; IL: interleukin; MCP-1: monocyte chemoattractant protein 1; MIP-1α: migration inhibitory protein 1-alpha; mtDNA: mitochondrial DNA; PAF: platelet-activating factor; PD-L1: programmed death ligand 1; PDGF: platelet-derived growth factor; PF4: platelet factor 4; PGE2: prostaglandin E2; RANTES: regulated upon activation and normal T cell expressed and secreted; S1P: sphingosine 1-phosphate; SDF-1: stromal-derived factor 1; TGF-β: transforming growth factor-beta; TNF-α: tumor necrosis factor-alpha; TRAIL: TNF-related apoptosis-inducing ligand; TSP-1: thrombospondin 1; TxA2: thromboxane A2; VEGF: vascular endothelial growth factor.

## 4. Discussion and Perspectives

The involvement of platelets in cancer progression through the modulation of tumor growth, angiogenesis, metastasis, and chemoresistance has been recognized for decades [354]. Tumor cells activate platelets, and platelets alter cancer cells and the TME and their interaction [64]. As a result, improving our understanding of platelets’ role in cancer would have a diagnostic, prognostic, and therapeutic impact.

Ovarian cancer is one of the many types of cancer in which platelets interact with immune and nonimmune cellular components of the TME, including endothelial cells, pericytes, CAFs, neutrophils, macrophages, and T and B lymphocytes. The adhesion molecules on the platelets mediate the platelets’ binding to endothelial cells and extravasation [355,356,357]. The proangiogenic factors released by activated platelets, including VEGF and angiopoietin-1, facilitate neovascularization, lymphangiogenesis [61], and remodeling of the ECM [100], required for tumor growth and metastasis. The binding of platelet CLEC-2 to podoplanin on pericytes [125] and the release of platelet-derived mediators, such as TGF-β and PDGF, promote pericyte accumulation and interaction with endothelial cells [64]. Thus, platelets, through pericytes, contribute to vessel density [127] and integrity [128,129]. Platelets [147,148], platelet microparticles [24], and platelet-derived mitochondria [24] contribute to the transformation of fibroblasts to CAFs, which promotes the peritoneal dissemination of ovarian cancer cells [160] and the formation of malignant ascites [358].

Platelets communicate with immune cells and induce a tolerogenic state [359]. Platelets bind neutrophils [181,182,183,188,189] and facilitate neutrophil recruitment to the TME. Cytokines and chemokines released from platelets promote neutrophil migration to the TME [360]. Neutrophils activate platelets and promote thrombosis by releasing prothrombotic mediators and generating NETs [360,361]. Platelet–macrophage interaction favors immunosuppressive macrophage polarization [222,362]. Activated monocytes, in turn, promote coagulation and thrombosis [230]. Platelets promote the exhaustion of CD4^+^ T helper and CD8^+^ cytotoxic T cells [321] in conjunction with Tregs and Bregs [363]. Moreover, platelets promote the expression of immune checkpoint molecules on ovarian cancer cells, which further supports cancer progression [14] and can affect patients’ responses to immunotherapies [307,364].

Thrombocytosis is responsible for the approximately 8-fold elevated risk of ovarian cancer occurrence and 5-fold mortality rate in ovarian cancer patients [28]. An elevated platelet count is significantly associated with poor progression-free and overall survival of patients with localized and advanced cancer stages [365]. Moreover, ovarian cancer patients with high platelet numbers respond less to chemotherapy and demonstrate shorter treatment-free intervals caused by increased resistance [366,367]. This makes platelet count an important parameter for the preoperative assessment of pelvic mass [368] and in evaluating the response to therapy in ovarian cancer patients [369].

Antiplatelet reagents that have been extensively studied in cardiovascular disease [370,371] can be considered for potential therapeutic goals in cancers, including ovarian cancer. Agents that hinder platelet functions, including aspirin, ADP receptor antagonists (e.g., ticagrelor and clopidogrel), thrombin receptor-targeting molecules, inhibitors of GP Ib-IX, P-selectin, and CLEC-2, can be used as complementary therapeutics in ovarian cancer. Although they have a role in preventing cancer progression [372,373,374], long-term usage of antiplatelet agents such as aspirin, clopidogrel, and ticagrelor is associated with increased bleeding risk [375,376]. The blockade of CLEC-2 can be a beneficial alternative for patients with elevated bleeding susceptibility, since this approach a lower tendency to cause hemorrhage due to its inhibition of platelet activation without substantially affecting aggregation [227,377]. Including agents that target EGFR, PDGFR, and FGFR in anti-VEGF therapy can also improve the efficacy of cancer treatment by rewiring multiple cellular pathways and minimizing drug resistance in ovarian and other cancers [89,378,379]. Apart from suppressing the viability, progression, and dissemination of tumor cells, platelet-targeting compounds interfere with platelet-induced angiogenesis, thrombosis, ascites formation, and immunosuppression. Multiple clinical trials have assessed the impact of antiplatelet agents, alone or with chemotherapeutic drugs, on preventing or treating ovarian cancer. Significantly, platelet counts or platelet-derived molecules can be used as diagnostic, prognostic, or predictive biomarkers [380].

Of note, thrombocytopenia, a side-effect of cancer treatment, is a common phenomenon in cancer patients [381]. Platelet transfusion and platelet-stimulating agent treatment are clinically exploited applications to recover the platelet counts of cancer patients with immune thrombocytopenia and those who have undergone chemotherapy or radiotherapy. On the other hand, thrombocytosis supports tumor growth and chemoresistance by enhancing platelet-directed alterations in cancer cells and TME components [30,382]. Therefore, extra caution must be taken during the treatment of thrombocytopenia to avoid promoting cancer progression.

## 5. Conclusions

Paraneoplastic thrombocytosis in ovarian cancer is due to IL-6 originating from cancer cells and consequent thrombopoietin released from the liver. Thrombocytosis and activated platelets can be detected in advanced diseases, and might be associated with cancer-induced venous thrombosis and reduced overall survival. Besides cancer cells, platelets interact with other components of the TME. Antiplatelet strategies might interfere with neoangiogenesis, lymphangiogenesis, ECM remodeling, and the immunosuppressive profile of the TME. In addition, antiplatelet agents may reduce the formation of malignant ascites and premetastatic niches. Future studies must examine platelet—TME interaction and plan specific treatments for ovarian and other cancers.

## Figures and Tables

**Figure 1 cancers-15-01282-f001:**
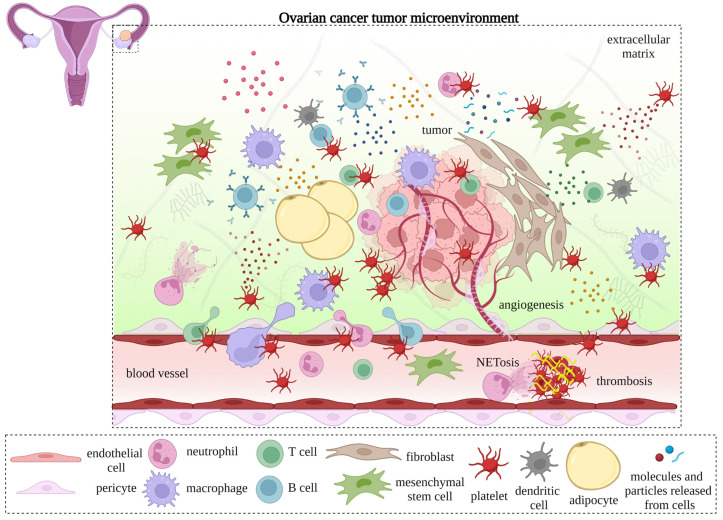
The tumor microenvironment (TME) of ovarian cancer. Tumor cell-induced activated platelets engage with endothelial cells, pericytes, mesenchymal stem cells (MSCs), cancer-associated fibroblasts (CAFs), adipocytes, immune cells, and extracellular matrix (ECM) elements through direct interaction or by releasing various modulatory factors and platelet microparticles. Platelets promote the extravasation, differentiation, and activation of these cells, which, in turn, contribute to the hyper-responsiveness of platelets and thrombosis in a positive feedback loop.

**Figure 2 cancers-15-01282-f002:**
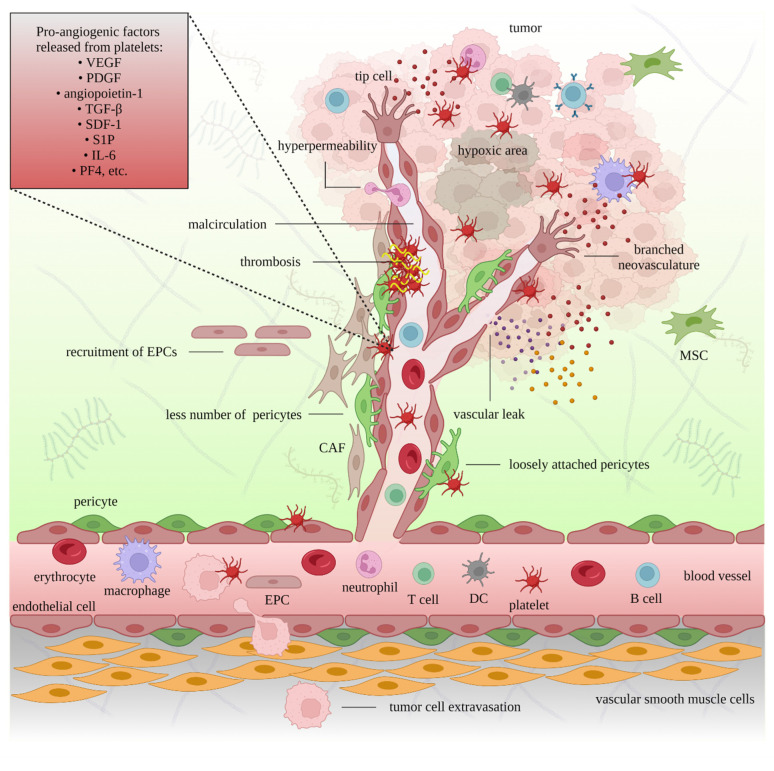
The pro-angiogenic role of platelets in cancer. The interaction between platelets, endothelial cells, and pericytes supports the extravasation of immune cells, mesenchymal stem cells (MSCs), endothelial precursor cells (EPCs), and tumor cells. Platelets also release proangiogenic factors that promote new blood vessel formation and facilitate tumor growth. The newly formed cancer-associated blood vessels are branched, leaky, and less supported by pericytes. Insufficient oxygen and nutrient supplies lead to hypoxic and necrotic areas in the tumor. Platelet-targeting strategies can restrict neoangiogenesis.

**Figure 3 cancers-15-01282-f003:**
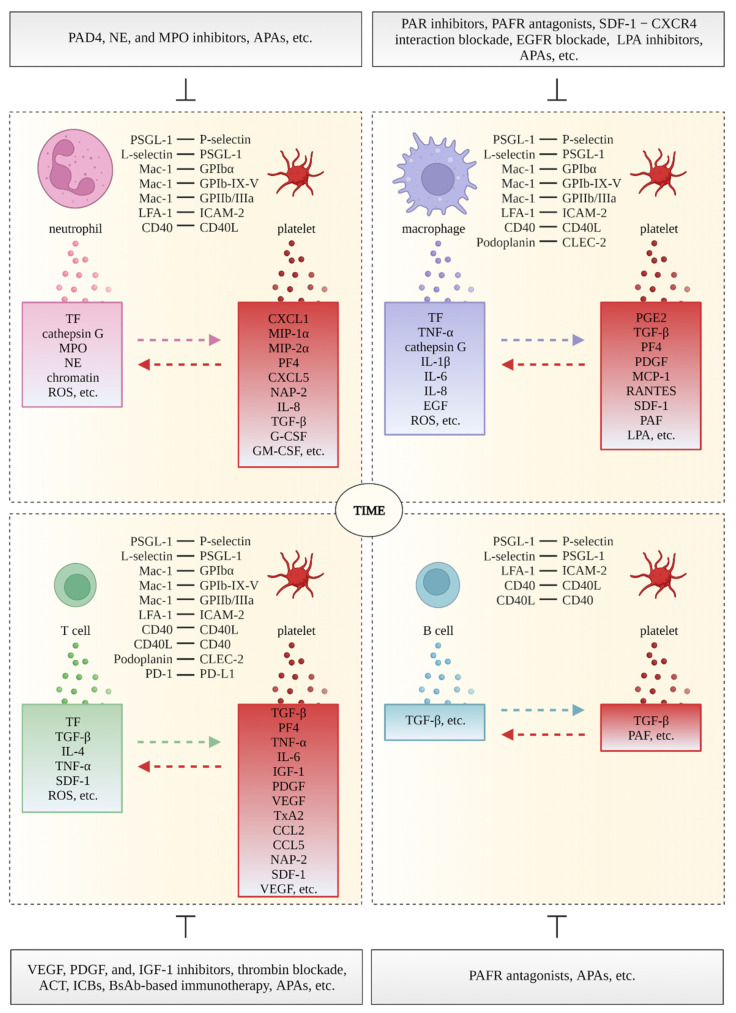
The molecules that mediate the interaction between platelets and immune cells in the tumor immune microenvironment (TIME). Activated platelets and immune cells in the TME communicate through various receptors and ligands or secreted modulatory factors, and these interactions can be targeted for the therapy of ovarian cancer.

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
