# Peer review of "Interactions between Platelets and Tumor Microenvironment Components in Ovarian Cancer and Their Implications for Treatment and Clinical Outcomes"

_cancers, 2023, doi:10.3390/cancers15041282_

Round 1
Reviewer 1 Report
I am a clinician-researcher, mostly involved in cancer diagnostics, and the relevance of platelets in that arena. Therefore, I am not strong in the mechanisms behind platelets' impacts on tumour growth/metastasis, etc. To some extent this makes me a good critical reader, though in other ways it makes me a poor critic (as I wouldn't spot errors).
My overall impression was of a comprehensive, well-structured and probably complete review (the word 'probably' reflects my ignorance, not an attempt to impugn the authors).
I was struck by the fact that platelets (and their various accompanying factors) all seemed to worsen the prognosis. Indeed, this extensive 'rap sheet' for platelets made me worry that the review was omitting positive aspects for platelets in the TME. I think that worry was unfounded, but I have a proposed solution. This could be one of two things:
1) I could get no idea from the text of the size of the effect for each of the platelet-derived factors on the TME. Particularly as there were so many interactions it would be helpful for the reader to have a sort of 'league table' of the adverse effects with the most common (or most nasty) effect at the top. This may not be easy, so it may have to be three categories: effect common/nasty or both; effect uncommon/less nasty; effect minimal. Indeed a fourth category may be added; platelet effect helpful rather than nasty.
2) an alternative would be to amend Figure 3 to capture some of what I've proposed above.
This request reflects my position as a clinician. Antiplatelet drugs get a mention (esp in the discussion) but if platelets are really that nasty, how close are we to proposing antiplatelet drugs for all OC patients (or all thrombocytotic cancer patients, irrespective of the cancer site?).
One minor point: line 82, I think the authors meant 'thrombocytosis' rather than 'thrombosis'.
But overall, this seems very comprehensive and is certainly well-written.
Reviewer 2 Report
excellen review. no suggestions
Reviewer 3 Report
This is an excellent paper with lots of insight and very informative figures, but...it's incomplete. The title announces a review that includes implications for ovarian cancer (OC) "treatment and clinical outcomes", but these aspects are reduced to a few lines (729-732). The authors mainly focus on pathophysiological aspects. Since one of the authors has co-authored some groundbreaking papers with a clear relevance to OC treatment, I cannot understand that these outstanding results (e.g. Bottsford-Miller et al. 2015, PMID: 25473001) are reported only marginally in 5 words (line 9), although they deserve a thorough discussion. Similarly, the paper does not adequately address the role of platelets in the diagnosis of OC (Watrowski et al. 2016 PMID:27207344) and prognosis of OC (Ye et al. 2019, PMID:30479089; Canzler et al. 2020, PMID: 32277253) as well as - more detailed - in the prediction of chemoresistance (Nakao et al. 2020, PMID: 32375852) or the monitoring of therapy response and disease recurrence of OC (Hu et al. 2020, PMID: 32682445). These are the implications for treatment and clinical outcomes. I would like to encourage the authors - recognized experts in this field - to discuss these aspects and publications in the form of an additional paragraph or two.
